Religious attendance after elevated depressive symptoms: is selection bias at work?

Balbuena Lloyd lloyd.balbuena@gmail.com
Baetz Marilyn
Bowen Rudy
Department of Psychiatry, University of Saskatchewan , Saskatoon, Saskatchewan , Canada
Moss Timothy
Electronic publication date: 2014 Mar 13
Publication date: 2014
Volume: 2
Electronic Location ID: e311
Received 2013 Dec 16; Accepted 2014 Feb 26
Copyright: © 2014 Balbuena et al.
Copyright year: 2014
Copyright holder: Balbuena et al.
License: This is an open access article distributed under the terms of the Creative Commons Attribution License, which permits unrestricted use, distribution, and reproduction in any medium, provided the original author and source are credited.
License URL: https://creativecommons.org/licenses/by/3.0/

Keywords: Selection bias, Religious attendance, Mental health

Funding: The authors received no funding for this work.

==============================
In an attempt to determine if selection bias could be a reason that religious attendance and depression are related, the predictive value of elevated depressive symptoms for a decrease in future attendance at religious services was examined in a longitudinal panel of 1,673 Dutch adults. Religious attendance was assessed yearly over five years using the single question, “how often do you attend religious gatherings nowadays?” Depressive symptoms were assessed four times within the first year using the Depression subscale of the Brief Symptom Inventory. Logistic regression models of change in attendance were created, stratifying by baseline attendance status. Attenders who developed elevated symptoms were less likely to subsequently decrease their attendance (relative risk ratio: 0.55, 95% CI [0.38–0.79]) relative to baseline as compared to those without elevated symptoms. This inverse association remained significant after controlling for health and demographic covariates, and when using multiply imputed data to account for attrition. Non-attenders were unlikely to start attending after elevated depressive symptoms. This study provides counter evidence against previous findings that church attenders are a self-selected healthier group.

Introduction

Attendance at religious services is usually reported to be inversely related to depression (Baetz et al., 2004; Balbuena, Baetz & Bowen, 2013; Hayward et al., 2012; Maselko, Gilman & Buka, 2009). However, as with many findings in the literature on religion and depression, the results have not been consistent, with other studies reporting a null (Ellison & Flannelly, 2009; Miller et al., 2012; Schnittker, 2001) or curvilinear relationship of attendance and depression (Taylor, Chatters & Nguyen, 2013). This inconsistency has been due in part to the predominance of cross-sectional designs, which cannot establish causation. We are aware of only three longitudinal studies examining the relation of depression to subsequent religious attendance. Recently, it was reported (Maselko et al., 2012) that women having an onset of depression prior to age 18 were more likely to stop attending religious services as adults compared to those with adult-onset MDE and those with no lifetime MDE. Similarly, a longitudinal study (Horowitz & Garber, 2003) reported that depressive episodes in grades 7–11 predicted lower levels of religious attendance in grade 12. The latter result is ambiguous because religious attendance in grade 6 also predicted lower odds of depression in grades 7–12, although the effect was just shy of statistical significance. Miller and colleagues (2002) followed a cohort of 146 individuals who had an MDE in childhood together with a control group of 123 who had no psychiatric disorder over 11 years. At follow-up, the rates of religious attendance between groups did not differ by childhood depression status. It is noteworthy that these studies covered the adolescence-to-adult transition period. Late adolescence is generally a time of profound change. The quest to establish one’s identity and career (Arnett, 2000) and experimentation with cohabitation, drugs, and alcohol (Benda & Corwyn, 1997; Thornton, Axinn & Hill, 1992; Uecker, Regnerus & Vaaler, 2007) could precipitate internal conflicts with religious doctrine, leading to declines in religious attendance. In short, the developmental processes occurring during adolescence confounds the relation of depression with religious attendance, in either causal direction.

Selection effects might work in two ways to confound the relation of depression with subsequent religious attendance. First, depressed individuals may withdraw from public worship as part of the overall social disengagement that occurs in depression (Maselko et al., 2012). Secondly, churchgoers might be less susceptible to depression. A meta-analysis of 94 studies reported a positive correlation of agreeableness, conscientiousness, and sociability with religious social investment (Lodi-Smith & Roberts, 2007). In summary, theory suggests that depressed individuals select themselves out of the churchgoing demographic while healthy ones select themselves into it. These selection effects, if operative, would likely overestimate the protective value of religion for mental health.

The protective value of religiosity does have empirical support. Coping with adversity through religion is well-documented in the literature (Pargament, 1997). “Turning to God or religion” as a coping strategy has been reported in cancer patients (Bussing, Ostermann & Koenig, 2007), newly bereaved individuals (Brown et al., 2004), and new immigrants (Connor, 2009). A study comparing religiosity and spirituality before and after HIV patients were informed of the diagnosis (Ironson, Stuetzle & Fletcher, 2006) reported that religiosity (including attendance) increased after the diagnosis became known. Furthermore, increased religiosity was associated with greater CD4 helper cells in the blood, indicating slower HIV progression (Ironson, Stuetzle & Fletcher, 2006). A US longitudinal study reported that depressed individuals were more likely to seek religious consolation—defined as searching for meaning in problems and difficulties (Ferraro & Kelley-Moore, 2000). Importantly, even those with no initial religious affiliation sought religious consolation after depression. This finding seems to support the “no atheists in foxholes” aphorism. However, adversity or trauma can also cause individuals to turn away from God or religion (Chen & Koenig, 2006; Fontana & Rosenheck, 2004). We are not aware of longitudinal studies reporting changes to religious attendance per se (vis-a-vis religiosity broadly speaking) after a depressive episode aside from the three studies in adolescence (Horowitz & Garber, 2003; Maselko et al., 2012; Miller et al., 2002) already mentioned.

In this paper, our main objective was to study whether elevated depressive symptoms introduce selection bias in a longitudinal follow-up of religious attendance. Our research questions were as follows. First, do religious attenders who develop elevated depressive symptoms subsequently decrease their attendance level? Second, do non-attenders who develop elevated symptoms begin to attend services?

Materials and Methods

Sample

The data is from the Longitudinal Internet Studies for the Social Sciences (LISS) panel, a random sample of adults living in the Netherlands (Scherpenzeel, 2011). The panel of almost 8,000 individuals was drawn from the Dutch-speaking population based on a list of addresses provided by Statistics Netherlands. The recruitment strategy is described in more detail elsewhere (Scherpenzeel, 2011). LISS is a continuing study in which participants complete online questionnaires monthly on topics including family, economic situation, health, and religion. Each LISS participant has a unique member ID, so it is possible to combine their responses across different survey modules and from one wave to another. For the purpose of our study, we merged the religion (n = 7, 418) and mental health (n = 1, 804) modules, narrowing our sample to those who responded to both surveys (n = 1, 718) (see Fig. 1A). Waves of religion and depression assessments were not simultaneous, so hereafter, use the notations Ri and Di to refer to religion and depression waves, respectively (see Fig. 1B). Our research question requires the following temporal sequence: R1 → D2 to D4 → R2. By comparing attendance level at R2 with R1, we can test whether depression predicts a relative decrease at R2. This design requirement forced us to exclude those individuals (n = 41) who had a different sequence of assessments:

D1 → D2 → R1 → D2 to D4 since for these individuals, one cannot compare attendance levels before and after depression. Our effective sample size was 1,673.

Figure 1 Subjects.

(A) LISS datasets used and participant flowchart; (B) schedule of religion and depression assessments.

In the sample, 887 individuals did not attend services and 786 attended services at R1. (Explained in Measures section below.) The characteristics of these groups are described in Table 1. After baseline assessment, D1, depression was assessed during a seven-month span from March to September 2008 while attendance was assessed another four times in addition to R1, and these assessments occurred on each January from 2009 to 2012. Our main analytic strategy was to compare the proportion of individuals that changed their attendance level, stratified by baseline attendance. The change in attendance level was indexed to R1 attendance since there were no further depression assessments after September 2008.

Table 1 Comparison of attenders and non-attenders.

Characteristics of Dutch individuals who responded to mental health (Wave 1) and religious attendance (Wave 1–Jan 2008) of the Longitudinal Internet Studies for the Social Sciences (LISS), Netherlands (n = 1, 673).

	Service attenders at R1	Non-attenders of service at R1	p of the difference	
n	786	887		
Mean age (sd)	49.60 (18.22)	45.90 (17.21)	<.001	
Civil status				
Unmarried	323 (41)	443 (50)	<.001	
Married	463 (59)	444 (50)		
Monthly income				
Low income (0–500 € per month)	77 (10)	65 (7)	.07	
Not low income (>500 € per month)	709 (90)	822 (93)		
Mean self-rated health (sd)*	3.12 (0.76)	3.13 (0.76)	.68	
With a chronic condition				
Yes	232 (32)	227 (27)		
No	497 (68)	600 (73)	.06	
Baseline distribution of BSI depression normative scores				
Very low	0 (0)	0 (0)	.38	
Low	0 (0)	0 (0)	
Below average	113 (14)	145 (16)	
Average	196 (25)	234 (26)	
Above average	284 (36)	324 (37)	
High	168 (21)	162 (18)	
Very high	25 (3)	22 (2)	
Notes.

Figures in this table are n (%) except for age and self-rated health which are mean (sd).

* Rated on a scale of 1–5, with higher scores indicating better health.

Measures

Religious attendance was assessed with the single question, “Aside from special occasions such as weddings and funerals, how often do you attend religious gatherings nowadays?” This was answered on a 7-point Likert scale coded as 1 = everyday, 2 = more than once a week, 3 = once a week, 4 = at least once a month, 5 = only on special religious days, 6 = less often, and 7 = never. We reverse coded the scale for ease of interpretation—i.e., larger number indicated higher attendance. Among attenders, we used a dichotomous attendance variable for decreased (coded 0) and same/increased (coded 1) and a categorical version: decreased, unchanged, and increased attendance. These categories were formed by subtracting the R1 attendance level from a given follow-up assessment. For non-attenders in R1, a dichotomous variable with unchanged (coded 0) and increased (coded 1) was used.

Our substantive predictor was assessed using the Depression subscale of the Brief Symptom Inventory (Derogatis, 1975). The subscale consists of 6 items rated on a 5-point scale from “Not at all” to “Extremely”. The BSI is itself a short version of the SCL-90 (Derogatis & Melisaratos, 1983). BSI Depression had an internal reliability ranging from 0.70 to 0.92 in four different studies (Boulet & Boss, 1991; Derogatis & Melisaratos, 1983; Johnson et al., 2008; Kellett et al., 2003) and correlated with the Beck Depression Inventory, r = .71 and r = .77, in two other studies (Stukenberg, Dura & Kiecolt-Glaser, 1990; Prinz et al., 2013). Two studies validated BSI Depression against either DSM-III or DSM-IV-TR major depression (Stukenberg, Dura & Kiecolt-Glaser, 1990; Johnson et al., 2008). ROC analyses reported area under the curve as 0.83 in community dwelling older adults, but a more modest 0.65 among psychiatric inpatients.

For copyright reasons, the LISS dataset only provided the normative category of each respondent and not the responses to individual questions. These normative categories of BSI depression were: 1 = very low, 2 = low, 3 = below average, 4 = average, and 5 = above average, 6 = high, 7 = very high. We dichotomized them into low (very low to above average) and elevated (high to very high). The cut-off was chosen because it resulted in a better approximation of the prevalence of depression in the Netherlands (Bijl, Ravelli & van Zessen, 1998) compared to alternative cut-offs.

Covariates

We controlled for possible confounders including chronic health conditions, income, marital status, age, and gender. Chronic conditions were assessed with the single question, “Do you suffer from any kind of long-standing disease, affliction or handicap, or do you suffer from the consequences of an accident?” Income was assessed using a monthly personal gross income with five categories: 0–500, 501–1500, 1501–2500, 2501–3500, and 3501 and higher Euros per month. Marital status was used as an index of social support and was categorized into 1 = Married; 2 = Separated/Divorced/Widowed; 3 = Not Married. All covariates were measured between December 2007 and January 2008.

Statistical analysis

We first compared demographic characteristics of attenders and non-attenders at baseline. To examine whether elevated depressive symptoms predicted subsequent decreased attendance, we crosstabulated dichotomized attendance with level of symptoms among attenders, for each religion follow-up year. Chi-square tests of association were performed. To examine change longitudinally and control for confounders, we created a multinomial logistic regression model among attenders. Relative risk ratios of decreasing or increasing attendance over maintaining attendance were calculated.

To examine if non-attenders start attending services after elevated depression, a binary logistic regression model was created with change in attendance (0 = no change; 1 = attended) as dependent variable. We entered the covariates in all regression models to control for confounding and robust standard errors were calculated to account for correlated errors in repeated observations.

Missing data

Missing values for the covariates at baseline did not differ between attenders and non-attenders. To examine whether our results would be biased by differential attrition between attenders and non-attenders, we performed multiple imputation on our dependent and predictor variables using a set of 15 socio-demographic and psychological variables assessed at baseline. (These variables are available from the authors upon request.) The imputation procedure was implemented using REALCOM-IMPUTE (Carpenter, Goldstein & Kenward, 2011) software and 10 imputed datasets were generated, which were then analyzed using Stata 12.1. Our regression models were then repeated using the multiply imputed data.

Results

Religious attenders were more likely to be female, older, and married as compared to non-attenders. Attenders and non-attenders were similar in self-rated health, in the proportion having chronic conditions, and in the distribution of BSI Depression normative scores. From March to September 2008, 491 individuals developed elevated depression. The chi-square tests in each of the four follow-up waves showed no association between depression status and subsequent decrease in attendance with one exception. In 2012, those with elevated BSI Depression among baseline attenders were more likely to keep or increase their 2008 attendance levels. In post-hoc analysis, we repeated the change in attendance analysis by depression category stratified first by gender and then by age category. In gender-stratified analysis, no associations were found in each attendance follow-up. In age-stratified analysis, no associations were found in the first two years after depression assessments took place. In both R3 and R4, those 48 years old and above and with elevated depression were more likely to maintain or increase attendance levels (both p’s < .05). (See Table 2. The tables stratified by gender or age category are available from the authors by request.)

Table 2 Change in attendance by depression status.

BSI depression vs changes in attendance in the Longitudinal Internet Studies for the Social Sciences (LISS), Netherlands.

Service attenders at R1	Non-attenders of service at R1	
2009 attendance vs 2008	Unelevated BSI depression n (%)	Elevated BSI depression n (%)	χ 2	p	2009 attendance vs 2008	Unelevated BSI depression n (%)	Elevated BSI depression n (%)	χ 2	p	
Same or increase	371 (73)	184 (78)	2.31	.13	Same	394 (67)	177 (69)	.47	.49	
Decrease	139 (27)	52 (22)			Increase	194 (33)	78 (31)			
2010 attendance vs 2008	Unelevated BSI depression n (%)	Elevated BSI depression n (%)	χ 2	p	2010 attendance vs 2008	Unelevated BSI depression n (%)	Elevated BSI depression n (%)	χ 2	p	
Same or increase	392 (77)	191 (81)	1.56	.21	Same	348 (59)	154 (60)	.11	.74	
Decrease	118 (23)	45 (19)			Increase	240 (41)	101 (40)			
2011 attendance vs 2008	Unelevated BSI depression n (%)	Elevated BSI depression n (%)	χ 2	p	2011 attendance vs 2008	Unelevated BSI depression n (%)	Elevated BSI depression n (%)	χ 2	p	
Same or increase	390 (76)	189 (80)	1.22	.27	Same	315 (54)	142 (56)	.32	.57	
Decrease	120 (24)	47 (20)			Increase	273 (46)	113 (44)			
2012 attendance vs 2008	Unelevated BSI depression n (%)	Elevated BSI depression n (%)	χ 2	p	2012 attendance vs 2008	Unelevated BSI depression n (%)	Elevated BSI depression n (%)	χ 2	p	
Same or increase	367 (72)	189 (80)	5.61	.02	Same	291 (49)	136 (53)	1.05	.31	
Decrease	143 (28)	47 (20)			Increase	297 (50)	119 (47)			
Notes.

If the individual had a norm-referenced score of “high” or “very high” in any of the three assessments in 2008, the individual was assigned to elevated and otherwise to unelevated.

BSI Brief Symptom Inventory

In multinomial logistic regression modeling, elevated symptoms predicted lower probability of a decrease in attendance (relative risk ratio = 0.55, 95% CI [0.38–0.79]) relative to baseline attendance levels among attenders (see Table 3 and Fig. 2). Among non-attenders, elevated symptoms were unrelated to a subsequent increase in attendance with one exception. The estimates were not materially different when the models were run using multiply imputed data.

Table 3 Regression models.

Logistic models of change in religious attendance levels over 5 years from the Longitudinal Internet Studies for the Social Sciences (LISS), Netherlands.

	Religious attenders at baseline	Non-attenders at baseline	
	Model 1a:
complete cases	Model 1Aa:
multiply imputed data
(10 imputations)	Model 2a:
complete cases	Model 2Aa:
multiply imputed data
(10 imputations)	
	Relative risk ratio
(95% CI)	Relative risk ratio
(95% CI)	Odds ratio
(95% CI)	Odds ratio
(95% CI)	
Outcome: decreased attendance	
Predictor: elevated depression	0.55 (0.38–0.79) **	0.58 (0.41–0.83) **	N/A	N/A	
Outcome: same attendance	(Reference category)	(Reference category)	
Outcome: increased attendance	
Predictor: elevated depression	0.90 (0.68–1.19)	1.27 (0.78–2.07)	0.93 (0.73–1.18)	1.10 (0.55–2.21)	
Notes.

a In these models, the following covariates have been controlled: existing chronic condition, gender, marital status, income, and age.

** p values: .01.

Figure 2 Probabilities.

Predicted probabilities of a decrease, same, and increase in religious attendance (95% CI).

Discussion

The main finding in this study is that elevated BSI depression does not cause a subsequent decrease in attendance. On the contrary, elevated BSI depression predicts same or increased levels of attendance as compared with prior to elevated BSI depression. None of our results across three analytical strategies showed a decrease in attendance. Although unstratified analyses did not yield a significant association, gender and age-stratified results indicated an association of continued or increased attendance with elevated symptoms. We do not interpret this association because other variables could possibly explain the association. The multinomial logistic model among attenders, with covariates controlled for, indicated that elevated symptoms predicted same or increased attendance. As a secondary finding, non-attenders were not likely to seek recourse in religion by starting to attend services.

Our main finding addresses the issue of selection bias in studies of religious attendance and mental health. To recap, when depressed individuals stop attending services, the apparent protection afforded by attendance would be inflated. This selection bias would be most pronounced in cross-sectional studies. In longitudinal epidemiologic studies, bias remains an issue because there might be differential attrition in attendance by depression status. Since we found that attenders who develop elevated symptoms become less likely to drop out of worship, then they might in effect be overrepresented among churchgoers. Therefore, the protective effect of religious attendance might in fact be underestimated, contrary to the reverse causation argument.

Religious individuals turn to their faith in times of distress as a means of coping (Brown et al., 2004; Bussing, Ostermann & Koenig, 2007; Ironson, Stuetzle & Fletcher, 2006). Religious worship is a source of social support and allows individuals to cope with life-threatening conditions by providing “medicine for the spirit” (Cummings & Pargament, 2010). A longitudinal study (Li & Ferraro, 2005) on depression and volunteering among older adults reported that elevated depression at wave 2 predicted seeking out volunteering opportunities at wave 3. This was explained as a compensatory mechanism to alleviate negative affect, via social integration. These older adults had physical health problems in addition to depression and the physical health problems were the operative barriers. Consistent with this finding, Chen & Koenig (2006) reported that in elderly, medically ill individuals, an increased severity of physical illness predicted a decrease in organizational religiousness. Importantly, the association was completely mediated by physical activity limitations. Together with our own results, these suggest that it is not depression per se that causes a disengagement from participation in religious service. In effect, reverse causation and non-random attrition might be more relevant to studies examining religious attendance and physical health.

The null association of elevated depression with subsequent attendance among non-attenders indicates that non-attenders do not start attending in the face of adversity. By contrast, Ferraro & Kelley-Moore (2000) reported that depression, cancer, and chronic conditions lead to consolation-seeking even among those with no religious affiliation. The two points of view can be reconciled because attendance is not necessary for consolation-seeking. The need to find meaning in illness is shared by believers and non-believers who may turn to nature, arts, music, and relationships (Burnard, 1988).

An intriguing study (Farias et al., 2013) reported that rowers about to enter a competition reported greater belief in science as compared with those in a training session. The authors concluded that affirming one’s secular worldview serves a similar function as religious faith in moments of stress or existential anxiety. Hence, affiliated individuals cope by means of religion while non-attending ones seek support and meaning elsewhere. It appears that depressive symptoms are not a strong driver of entry into or exit from the churchgoing demographic. More empirical research is needed to validate this interpretation.

Our study has several important limitations. First, we relied on the depression subscale of the BSI to assess depressive symptoms. It has been argued that although the global BSI scale is a valid measure of distress, it is not intended to screen for particular psychiatric disorders (Asner-Self, Schreiber & Marotta, 2006). We set the BSI cut-off for depression at “high” or “very high” but there is no way to tell whether this is in concordance with clinically diagnosed depression. We therefore allow for the possibility that the depressed group that we studied might differ from a group with formally diagnosed major depression (Sareen et al., 2013). Secondly, there were no assessments of depression from 2009 to 2012—the period during which religious attendance was compared to initial attendance. As such, all religious attendance changes were indexed to depression levels in 2008 only. We cannot rule out the possibility that attendance levels after 2008 were related to unmeasured elevated symptoms after 2008. Third, the sample that we worked with was restricted to LISS participants who responded to both religion and mental health modules and might not be representative of the Dutch national population. Fourth, while we have ruled out sampling bias in terms of attrition in attendance, we could not address whether personality traits such as greater resilience to depression characterized attenders in the first place. Finally, other health-related reasons, including physical disability, might be correlated with depression and could cause selection bias. This is beyond the scope of our study. The major strength of this study was that depression assessments were proximal to those of religion. Our sample also consisted of a small fraction (5%) of late adolescents and is less affected by the well-reported decline in religious attendance in adolescence.

Within the limitations of our study, we conclude that elevated depressive symptoms do not cause religiously affiliated individuals to subsequently decrease attendance at religious services.

The authors are grateful to the CentERdata at Tilburg University in the Netherlands for making the data available for use.

Additional Information and Declarations

Competing Interests

Author Contributions

The authors have no financial or competing interests to declare.

Lloyd Balbuena conceived and designed the experiments, analyzed the data, wrote the paper, prepared figures and/or tables.

Marilyn Baetz and Rudy Bowen conceived and designed the experiments, wrote the paper, reviewed drafts of the paper.

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
