# Peer review of "Religious attendance after elevated depressive symptoms: is selection bias at work?"

_PeerJ, doi:10.7717/peerj.311_

## Round 0.1 · original submission · Major Revisions

Thank you for your submission of this paper to PeerJ. It has now been possible to obtain independent peer reviews for the work. Thank you for your patience while this was in process.

As you will see from the reviewers comments, there are clearly differing views as to the merits of the paper, largely on technical grounds. One of the reviewers was essentially positive about the work, identifying only minor features in the basic reporting as potential issues to be addressed. However, the other reviewer identified what may be more serious technical flaws in the interpretation and analysis of the data presented. In particular, I would draw your attention to the issues raised around timescales in baseline/follow-up, and also the sampling issue around those who did not answer the depression module. Other issues raised by this reviewer might be addressed by re-presentation of some of the data and rewording of some of the interpretation and considering carefully the issues of selection bias.

I would like to give you the opportunity to make revisions to the paper to address the issues identified by the reviewers and to resubmit the work for reconsideration. If you choose to resubmit, please include in your cover letter a careful point by point description of your response to issues raised in review. Please note however that these issues are not considered trivial, and that I cannot give assurance as to any decision that may be made about a resubmission. Nevertheless, I do hope to have the opportunity to see a revised manuscript.

Reviewer 1 ·

Basic reporting

This is a very well written and clearly set out article. The one place where a word is misused that I could find is on line 230, where the word 'paradoxically' is used to mean 'contrary to this', and I would suggest that this is changed.
I would also question the strength of the assertion in line 54. It would be more appropriately advanced as a hypothesis - social activities including religious gatherings may be less rewarding during depressive episodes. I also think the reference to the amgydala study is fairly arbitrary and would suggest that it is removed. After all a core defining aspect of depression is loss of pleasure in things that under normal circumstances would provide pleasure.

Experimental design

Excellent. the statistical analysis is appropriate to the design and carried out competently.

Validity of the findings

Small effects, but the authors are very clear about the limitations of the study and its scope.

Additional comments

This is an important question that is addressed by a careful and appropriately designed empirical study. It starts to clarify some of the uncertainties in the temporal relationship between mental health and religious observance. It would be good to see more work by this group perhaps using qualitative methods to unpack further some of the processes that are at work when people turn to their faith community during times of elevated depressive symptoms.

·

Basic reporting

no comments

Experimental design

no comments

Validity of the findings

The most significant weakness of the analysis is the sequence of the depression and attendance measurements vs. their use in the analysis. For example, depression data collected in March 2008 is treated as a ‘follow-up’ to the ‘baseline’ attendance data from April 2008. Similarly, the ‘baseline’ data on attendance and depression that is separated by 4 months (December to April) is treated as effectively simultaneous, although the 2 month period from April 2008 to June 2008 (ignoring the March 2008 issue) is treated as a longitudinal follow-up.

Who are the people who did not answer the depression module? It is important to note whether they did not receive the module or chose not to answer it.

The exclusion of 419 depressed people in 2007 (a substantial portion of the sample) introduces a bias – the sample in the analysis is by definition healthier, although it's not known whether they have never been depressed.

Additional comments

There seems to be some confusion about the meaning of selection bias or reverse causation. For example, In the paragraph beginning with line 61, the authors follow the statement that ‘the protective value of religiosity also has empirical support’ with examples of how individuals turn to religion when faced with a health problem. This is also an example of reverse causation since health precedes religion, although in these cases, religiosity is increased (vs. decreased) as a result of a health problem. Importantly, this is the very finding that the authors report - that those with depression symptoms are more likely to increase attendance compared to those without. In the current dataset, this kind of selection would 'wash out' any positive causal beneficial impact of attendance - exactly what selection bias does.

In table 2, it is not clear why the 2009, 2010, 2011 and 2012 attendance is separated out in this table since it is effectively collapsed in the regression analyses. There is clearly a secular trend of decreasing attendance over time – are the authors suggesting that, for example, new depression symptoms in 2008 would impact attendance in 2012 but not 2008? If the authors choose to keep the table, for balance, if there is a bivariate table (Table 2) for baseline attenders, similar information should be presented for the non-attenders at baseline, or reasons for not presenting that data should be given.

Table 3 presentation is very confusing - if the outcome is labeled as ‘decreased attendance’ why would this be the reference category?

The use of relative risk ratios is somewhat unusual and so it would be helpful for most readers to explain that this is not the same as a risk ratio or a relative risk (and is more like an odds ratio).

---

## Round 0.2 · accepted · Accept

In my opinion, and that of independent re-review, the manuscript is much improved and clarified. The considerable trouble that you have taken with this revised version is evident, and the paper significantly enhanced on the basis of it.

·

Basic reporting

The paper reads much, much better. Although I realize the authors were concerned by the double negative phrasing of the results, they are now more easy to follow.

Experimental design

no comments

Validity of the findings

no comments

Additional comments

The authors have addressed my major concerns.